# Active Breaks Reduce Back Overload during Prolonged Sitting: Ergonomic Analysis with Infrared Thermography

**DOI:** 10.3390/jcm13113178

**Published:** 2024-05-29

**Authors:** Martina Sortino, Bruno Trovato, Marta Zanghì, Federico Roggio, Giuseppe Musumeci

**Affiliations:** 1Department of Biomedical and Biotechnological Sciences, Section of Anatomy, Histology and Movement Science, School of Medicine, University of Catania, 95123 Catania, Italy; martina.sortino@phd.unict.it (M.S.); bruno.trovato@phd.unict.it (B.T.); marta.zanghi@phd.unict.it (M.Z.); g.musumeci@unict.it (G.M.); 2Research Center on Motor Activities (CRAM), University of Catania, 95123 Catania, Italy; 3Department of Biology, Sbarro Institute for Cancer Research and Molecular Medicine, College of Science and Technology, Temple University, Philadelphia, PA 19122, USA

**Keywords:** active breaks, infrared thermography, musculoskeletal disorders, office workers, physical exercise, prevention, prolonged sitting, workplace interventions

## Abstract

**Background:** Prolonged sitting is a potential risk factor for musculoskeletal disorders in office workers. This study aims to evaluate the effect of active breaks on reducing muscle overload in subjects who sit for long periods using infrared thermography (IRT). **Methods:** A sample of 57 office workers participated in this study and were divided into two groups: active breaks (ABs) and no active breaks (NABs). The NAB group sat continuously for 90 min without standing up, while the AB group performed stretching and mobility exercises every 30 min. IRT measurements were taken every 30 min before the active breaks. **Results:** The results highlight that the skin temperature of the back increased significantly in both groups after 30 min of sitting; however, in the subsequent measurements, the AB group showed a decrease in temperature, while the NAB group maintained a high temperature. Exercise and time point of measurement all reported *p*-values < 0.001; there were no statistically significant differences between the Δ_t0-1_ of the NAB and AB groups, while the Δ_t1-2_ and Δ_t1-3_ of the NAB and AB groups showed statistically significant differences for all back regions. **Conclusions:** The clinical relevance of this study confirms the negative effects of prolonged sitting on the health of the back, demonstrating that active breaks can reduce back strain, emphasizing the need for workplace interventions. In addition, IRT represents a non-invasive method to assess back muscle overload and monitor the effectiveness of interventions in all categories of workers who maintain a prolonged sitting position. The main limitation of this study is the absence of a questionnaire for the assessment of back pain, which does not allow a direct correlation between temperature changes and back pain outcomes.

## 1. Introduction

Body temperature is the outcome of the ability of the human body to produce and release heat through the mechanism of thermoregulation, an activity necessary for vital metabolic functions. While core body temperature fluctuations often indicate systemic inflammation or pathologies, variations in skin temperature (Tsk) distribution can reveal different health concerns. This field has been transformed by infrared thermography (IRT), as it provides a non-invasive and rapid method for detecting body temperature, although it does have some particular characteristics that need to be considered for reliable measurements [1]. IRT cameras, as highlighted, use an advanced radiation thermometer to detect the thermal emissions of a body in an electromagnetic spectrum invisible to human eyes [2]. These cameras measure the surface temperature to collect information about blood flow and muscle metabolism, both of which improve heat convection throughout the body [3]. The Tsk of the region of interest (ROI) framed by the camera is the outcome measure most reported in the literature and evaluated with this technique [4,5]. This method is simple and safe but also highly reproducible, non-invasive, and requires no physical contact with the patient [6]. The attention of researchers towards the use of IRT as an auxiliary tool in the analysis of the human body is increasing, particularly in the application of the musculoskeletal system [7,8] within different age ranges [9] or daily activities [10]. Moreover, the application has also extended to other fields, such as sport [11] and posture [12], but also in pathological subjects with metabolic alterations [13], breast cancer [14], rheumatic diseases, and osteoarthritis [15,16], and even in the field of musculoskeletal disorders [17]. IRT has been applied to various fields of musculoskeletal disorders, such as scoliosis [18], arthritis [19], and low back pain [20]. The interest in this method has increased and is being invested in many fields. It has been seen, for example, that an increase in temperature relative to the basal one correlates with muscle fatigue [21]. Moreover, increased skin temperature has also been observed in sports as a symptom or synonym of overload and potential injuries, suggesting that infrared thermography can be an effective tool to monitor athletes’ health and prevent injuries [22].

In the current century, a sedentary lifestyle has emerged as a major contributor to musculoskeletal changes that lead to acute and chronic pain [23], and it is a significant risk factor for various cardiovascular diseases and metabolic disorders [24]. In addition, the global burden of diseases report shows that among young adults, musculoskeletal diseases are the main causes of disease, leading to many years of disability [25]. Prolonged sitting poses a potential health risk to the musculoskeletal system of employees [26]. By monitoring changes in skin temperature across working muscles, IRT can help identify potential ergonomic risks and areas for improvement within the work environment [27]. While IRT provides an indirect measure of the thermal radiation of the body, it offers valuable insights into how the workplace impacts the body and potential areas of overload. This information can be used to suggest modifications in equipment, work processes, or the environment, ultimately aiming to reduce risks and enhance worker well-being [28]. Some researchers observed an increased discomfort in the lower back following long periods in a sitting position. One study that analyzed subjects working at a computer found that more than half of the participants experienced musculoskeletal complaints in the neck and upper limb area within one year [29]. The health risks of office workers are a growing concern for industries and societies as sedentary work becomes more prevalent [30]. Consequently, it is crucial to understand the impact of long sessions of sitting on musculoskeletal health, particularly of the back.

It is clinically relevant to consider short and active breaks to contrast these risks by improving blood circulation, increasing energy expenditure, and reducing muscle tension. Active breaks are one potential strategy to address this problem, and indeed are essential for maintaining good back health for those who spend a lot of time sitting at work, as these reduce the risk of musculoskeletal disorders and strengthen and lengthen the back muscles [31,32]. During these breaks, moving and light stretching or walking can help release tension in the back muscles, correct posture, and promote overall spinal health. This proactive approach promotes the endurance and flexibility of back muscles in the long term, as well as contributing to the immediate release of discomfort [33]. Given the significant potential of IRT and active breaks in the prevention of musculoskeletal disorders, transitioning them into the working environment becomes crucial.

The prevalence of musculoskeletal pain is high among sedentary workers whose jobs require prolonged sitting [34]. Office workers, teachers, and drivers are particularly vulnerable to a range of musculoskeletal disorders due to the static and slouched posture maintained for hours [35]. Muscles become tense and stiff, joints lose flexibility, and improper posture stresses the spine and the other body joints. The sedentary nature of these occupations, coupled with the lack of breaks for movement or stretching, exacerbates the risk of developing chronic musculoskeletal discomfort. The hypothesis of this study is that administering quick and easy stretching and mobility exercises during working hours can alleviate the back overload caused by prolonged sitting and delay the onset of muscle stiffness, reducing the risk of musculoskeletal disorders.

Therefore, this study aims to assess the thermal variations in the back caused by extended periods of sitting and to explore how active breaks can reduce the burden of the back in office workers through the IRT analysis of the back during prolonged sitting and active breaks.

## 2. Materials and Methods

This observational study involved 57 office workers, 26 women and 31 men, aged between 30 and 40 years, to analyze thermographic changes in their backs during working hours. The participants were recruited at the Research Center on Motor Activities (CRAMs) of the University of Catania, Italy. The exclusion criteria were: musculoskeletal disorders, spinal pathologies, back dysmorphisms, and episodes of low back pain during the previous 4 months. The study was conducted following the Declaration of Helsinki and approved by the Institutional Review Board of the Research Center on Motor Activities (Protocol n.: CRAM-42-2024, 29 January 2024). All participants completed informed consent before taking part in the study. A first thermal assessment of the spine was carried out using the FLIR E54 thermal imaging camera (Wilsonville, OR, USA) according to the “Thermographic Imaging in Sports and Exercise Medicine (TISEM)” guidelines [4]. All participants were given a 15 min rest before taking the first IR image for acclimatization. The subjects were asked to wear a loose-fitting T-shirt, and to remove it at each thermographic measurement, remaining with the back naked. The participants were divided into two groups: the active break (AB) group and the no active break (NAB) group. The NAB group was seated for a continuous period of 90 min without getting up. In contrast, every 30 min, the AB group engaged in stretching and mobility exercises. Data were collected at the beginning (T0), after 30 min (T1), after 60 min (T2), and at the end of 90 min (T3). While the individuals in the NAB group stood up only for thermographic measurements, those in the AB group stood up both for measurements and to immediately perform their exercise routine, Figure 1.

### 2.1. Physical Activity Protocol

The participants were asked to perform a series of exercises that included mobility of the spine, stretching of the neck and back muscles, and to perform a short walk. The whole protocol lasted approximately 2 min, and the exercises were: walking for 20 m, remaining in trunk flexion for 20 s, 10 trunk rotations, 10 lateral trunk flexions, and 5 lateral neck extensions per side (Figure 2). The experimental protocol administered to the AB group is shown in Figure 3.

### 2.2. Infrared Thermography

All thermographic measurements were taken with the FLIR E54 thermal imaging camera (Wilsonville, OR, USA). This thermal camera is characterized by a resolution of 320 × 240 pixels and a thermal sensitivity of less than 0.04 °C. For the measurement set-up, the camera was set up on a tripod positioned at 1.5 m from the subject. Measurements were carried out in an environment with a constant temperature of approximately 23 °C ± 2 °C and a relative humidity level of 50%. The camera’s emissivity setting was set to 0.98. To analyze the thermal data, three ROIs were marked using FLIR Thermal Studio Pro^®^ software 1.9.38.0, focusing on specific areas of the spine: the cervical, dorsal, and lumbar regions. These ROIs were defined concerning the vertebrae of the spine, deliberately excluding the upper limbs. The cervical area was mapped as a rectangular section extending from the third to the seventh cervical vertebrae. The dorsal section of the body was outlined as a polygonal shape extending from the first to the twelfth thoracic vertebrae. The lumbar region was defined with a polygon extending from the first to the fifth lumbar vertebrae (Figure 4).

### 2.3. Statistical Analysis

The data analysis was conducted using R Project 4.3.2 for Statistical Computing (Vienna, Austria). The data were initially screened to ensure adherence to the assumptions of the parametric tests using the Shapiro–Wilk test for normality and Levene’s test for the homogeneity of variance. To investigate the influence of the exercise administration (yes/no), back area (cervical, dorsal, and lumbar), and measurement time points (T0 to T3) on the temperature, a mixed ANOVA was conducted. This analysis aimed to explore potential interactions between these factors. Follow-up analyses examined the main effects of the exercises and measurement time points on the temperature as well as their potential interaction. Since only specific interactions were deemed theoretically important a priori, post-hoc investigations were focused on those comparisons. Pairwise *t*-tests with the Benjamini–Hochberg *p*-value adjustment method were employed to control the family-wise error rate, allowing for targeted comparisons between exercise types and measurement points.

Furthermore, to specifically assess temperature changes following exercise administration (T1), temperature differentials (Δt) were calculated between T1 and the other time points (T0, T2, and T3). The same analysis pipeline used for the initial temperature analyses was then applied to highlight how patterns of change might differ by exercise condition and back area.

## 3. Results

The anthropometric measurements are shown in Table 1, while the mean temperature values for each back area in the exercise group (AB) and no exercise group (NAB) are reported in Table 2. The mixed ANOVA revealed a statistically significant difference between the two groups. The exercise, back area, and measure time points all reported *p*-values < 0.001; the combined effect of the exercise and measure time points also yielded a *p*-value < 0.001. We subsequently analyzed each back area (cervical, dorsal, and lumbar) individually. All areas demonstrated a *p*-value < 0.001 for the interaction between exercise and measure time points. A post-hoc analysis with Benjamini–Hochberg *p*-value adjustment was used to assess the significance of specific interactions in the AB and NAB groups, as outlined in Table 3. As hypothesized, temperature changes over time were observed in both groups. The cervical area of the NAB group showed a statistically significant temperature increase between T0 and T1, with no subsequent decrease in temperature. This pattern held true for the dorsal and lumbar areas, where only T0 to T1 comparisons were statistically significant. All other NAB group comparisons were not significant. In contrast, the back areas of the AB group showed statistically significant differences between all the time points, except for T1 to T2 in the dorsal area, which was slightly above the *p*-value threshold (*p* = 0.052). These differences indicate an initial temperature increase at T1 followed by a decrease in the subsequent time points for the AB group.

For the temperature differential, we followed the same statistical methodology to analyze the temperature differential between T1 (the start of exercise administration) and the subsequent time points. The mean temperature differential values for each back area in the AB and NAB groups are reported in Table 4 and Figure 5. The mixed ANOVA revealed a statistically significant difference between the groups. The exercise, back area, and measure time points reported *p*-values < 0.001; the combined effect of the exercise and measure time points also yielded a *p*-value < 0.001. The individual analysis of the back areas showed a *p*-value < 0.001 for the interaction between exercise and measure time points in the cervical and lumbar areas. The dorsal area yielded a *p*-value of 0.073. Post-hoc analyses clarified specific interaction differences and are reported in Table 5. A consistent pattern emerged across all areas: no statistically significant difference existed between Δ_t0-1_ of the NAB and AB groups. However, Δ_t1-2_ and Δ_t1-3_ of the NAB and AB groups demonstrated statistically significant differences. This indicates that the groups did not differ in temperature differential prior to exercise but showed significant differences after exercise administration.

## 4. Discussion

Prolonged sitting poses a serious health risk not only to office workers but to all categories of people who adopt this habit due to the demands of their work. This constant pressure leads to muscle overload and often debilitating pain. To face this problem, it is essential to incorporate regular movement and exercise breaks throughout the workday to mitigate the burden on the back. To the best of our knowledge this is the first study that investigates the effects of active breaks on prolonged sitting with infrared thermography. We examined the effect of active breaks with exercise on back temperature during prolonged sitting by comparing a group of office workers who performed exercise (AB) with one who did not perform it (NAB). A significant impact of exercise on Tsk was found, with variations depending on the area of the back examined and the time point evaluation.

Our observations show that back strain tends to increase after 30 min of sitting (Figure 6). However, performing specific exercises led to a reduction in back temperature after an additional 30 min. In contrast, we observed a sustained elevation in back temperature in those not performing exercises. Specifically, a temperature increase was observed in both groups from T0 to T1; however, at subsequent measurements (T2-T3), the two groups performed differently. The AB group showed a decrease in back temperature in all the three ROIs. Of these, only the dorsal area showed a threshold slightly above significance (*p* = 0.052), while the NAB group maintained a consistently high temperature like T1.

In our sample, maintaining a static sitting posture for 30 min elicited a significant increase in the Tsk of the cervical, dorsal, and lumbar ROIs. Prolonged sitting often leads to frequent trunk adjustments, which can put varying loads on the spine. A previous investigation found that office workers change their lumbopelvic position about every 6 min [36]. This muscle activity can be indirectly observed with infrared thermography [37], which can reveal a temperature increase associated with muscle tension [38]. A study on the ergonomics of sitting in office workers stated a correlation between the increase in fatigue and muscle stiffness during prolonged sitting [39]. We observed that condition within our sample since both groups showed an increase in the mean temperature of all the ROIs after 30 min of static sitting. We can infer that prolonged sitting can lead to muscle tension and fatigue [33], which might result in an increase in Tsk [40].

While office worker discomfort stems from various psychological, social, and environmental factors [41], prolonged sitting poses a serious postural risk for neck and lower back pain. This posture creates a sustained load on the spinal muscles, potentially leading to fatigue and tension [42]. This may be consistent with the theory that a change in Tsk may reflect a change in overload activity [43] with muscle tension related to sitting posture. Concerning the temperature changes, in the AB group, the mean temperature dropped significantly between T1-T2 and T2-T3 in the cervical and lumbar areas. This result suggests that active breaks may be effective in reducing muscle overload and fatigue induced by static sitting. Implementing active breaks in the workplace can improve the physical state of the body, keeping employee productivity levels high without causing any adverse effects [44].

To understand where the temperature difference occurred, we analyzed the differentials between specific time points. Our analysis of temperature differentials between T1 and the subsequent time points revealed a consistent pattern across all areas. Initially, there were no significant differences in Δ_t0-1_ of the NAB and AB groups. However, significant differences emerged in Δ_t1-2_ and Δ_t1-3_. This indicates a similar initial temperature increase in both groups, followed by a sustained temperature in the NAB group and a temperature reduction in the AB group. We associate this reduction with a positive response to the administered treatment. Our results align with previous studies demonstrating that active break programs reduce back discomfort [45,46,47]. Importantly, our infrared thermography approach provided a more detailed picture, revealing thermal changes in the back in response to the active break plan. Since infrared thermography is an easy, quick, safe, non-invasive, and objective method to assess these changes, our results are objective and not influenced by the subjective perceptions of individuals, unlike what might happen with questionnaires. Considering the findings of our study, where the AB group showed a decrease in temperature in comparison to the NAB group, we hypothesize that one of the underlying reasons behind the positive effect of active breaks may be the reduction of excessive muscle activation, resulting in a consequent decrease in temperature and not remaining increased in a plateau phase, as is the case in the NAB group.

There is a specific condition to consider: participants were instructed to maintain a static seated position for 30 min without standing. While this aligns with our research design, we recognize it differs from the typical routine of an office worker, where they might stand or move to perform various tasks. This controlled protocol ensured the specific exercises we administered were the primary factor, something that cannot be guaranteed in a standard workday. We are aware of a limitation of our study, which is the lack of a questionnaire to assess back pain, preventing us from directly correlating temperature changes with back pain outcomes.

The clinical relevance of this article reinforces the growing interest in improving the well-being and general health of those who perform sedentary work, as these categories of people often experience discomfort and back pain due to a lack of mobility. These findings are in line with previous research exploiting the harmful effects of prolonged sitting on the health of the back [48]. Our results emphasize the need for preventive measures in the workplace to reduce the risk of developing musculoskeletal problems typical of sedentary office work environments. Physical activity and ergonomic interventions can promote a healthier and more productive workplace for employees with beneficial effects also for companies. Research has widely shown that incorporating physical activity into the workday can lead to improved health and well-being for workers. According to some studies [49,50], introducing physical activity within the workday increased well-being, health, productivity, and mood and instead decreased absenteeism and sedentary behavior. These interventions can include changes at the individual, environmental, and organizational levels to increase physical activity or reduce sedentary behavior [51]. In addition, ergonomic interventions play a crucial role in improving the physical well-being of sedentary workers. Workstation adjustments and ergonomic training have been proposed to improve the sitting posture of office workers [48].

The results suggest that maintaining a static position during working hours might overload the back muscles, as observed using IRT. Our protocol also shows the possible positive effects of specific active breaks in reducing muscle overload and fatigue induced by static sitting. This study holds significant clinical relevance for office workers who are particularly susceptible to the detrimental effects of prolonged sedentary behavior. IRT offers a non-invasive and sensitive tool for detecting the early signs of muscle strain and overload, even before the worker experiences overt pain or discomfort, allowing for proactive interventions to prevent the development of more serious musculoskeletal disorders. By identifying specific areas of muscle overload, the results of this study can inform the design of targeted active break exercises that most effectively release tension and restore balance within the affected muscle groups. The simplicity of the active break protocol and its focus on short and manageable movements make it accessible for broad implementation in office settings, providing workers with an actionable tool to take control of their own health and mitigate the risks of sedentariness. The potential benefits extend beyond the office environment, with findings on the efficacy of active breaks having implications for any occupation involving prolonged sitting, including students, drivers, and those with limited mobility.

However, several limitations need to be acknowledged. Firstly, the relatively small sample size prevents us from generalizing the findings to the overall population. Secondly, while IRT is a valid ergonomic assessment tool, our reliance on it alone means the study lacks comparisons with other methods. Thirdly, requiring participants to remain seated without standing may not accurately reflect everyday working conditions. Finally, since we focused solely on office workers, their sedentary behavior could differ from that of teachers or drivers. Further studies are needed to confirm these results in other work environments or without specific indications. In addition, future studies may also investigate the correlations between back temperature, electromyographic activity, and back pain, and the mechanical properties of muscles like stiffness, tone, and elasticity, to gain a deeper understanding of how to improve sedentary workers’ daily living.

## 5. Conclusions

Prolonged sitting among office workers is a significant contributor to back overload and pain. Our study underscores this by showing that just 30 min of static sitting can notably raise back temperature, indicating a potential muscle overload. These findings emphasize the critical role of incorporating active breaks into the workday to ameliorate worker health. Administering a tailored physical exercise program may mitigate this muscle strain. This study identified a temperature plateau among those who did not perform the exercise program, contrasting with a decrease in back temperature observed in those who participated. These findings not only underline the utility and feasibility of using infrared thermography to assess back overload but also highlight the clear benefits of integrating an active break program for people who maintain a sitting position for a long period of time. The sample size, along with the sole use of infrared thermography and the request to maintain a seated posture, may represent a limitation of the study.

## Figures and Tables

**Figure 1 jcm-13-03178-f001:**
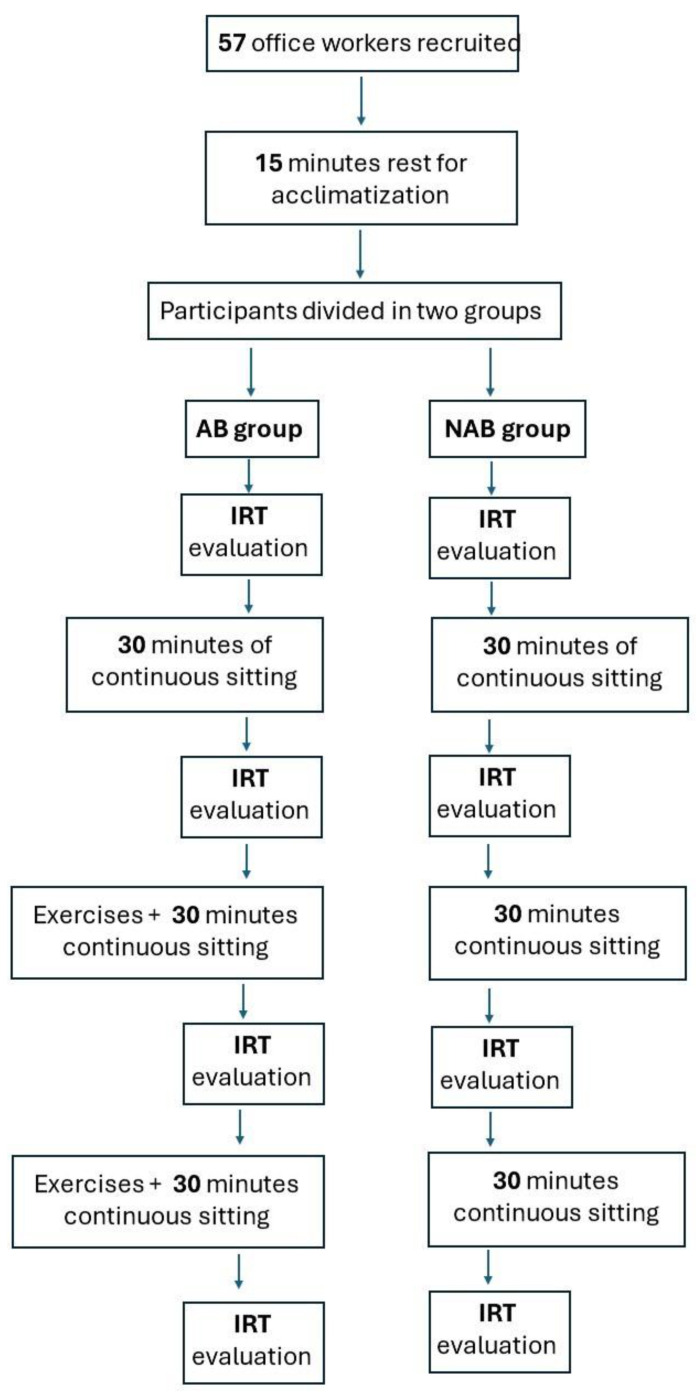
Flow diagram of the study protocol.

**Figure 2 jcm-13-03178-f002:**
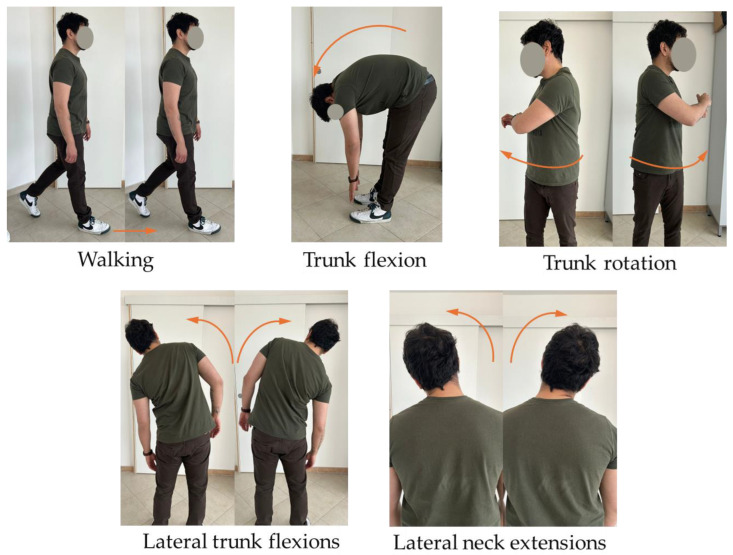
Physical activity protocol: walking for 20 m, remaining in trunk flexion for 20 s, 10 trunk rotations, 10 lateral trunk flexions, and 5 lateral neck extensions per side.

**Figure 3 jcm-13-03178-f003:**
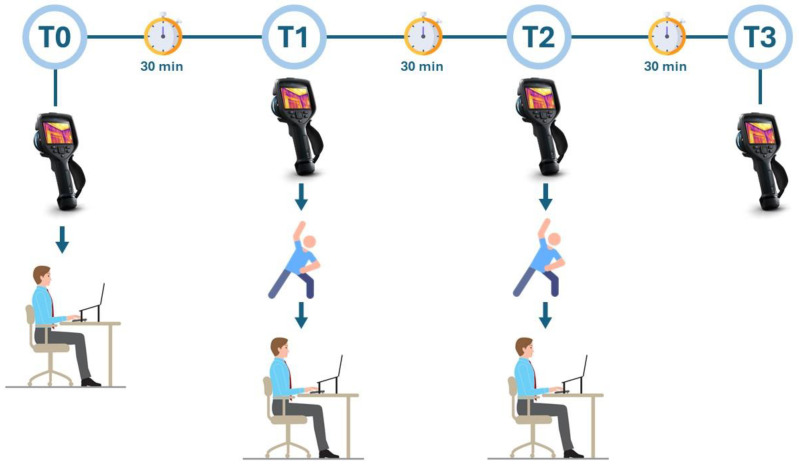
Experimental design of the study. IRT: infrared thermography measurement; T0: thermal evaluation at baseline; T1: thermal evaluation after 30 min of sitting and following the exercises; T2: thermal evaluation after 60 min of sitting and following the exercises; T3: thermal evaluation after 90 min of sitting.

**Figure 4 jcm-13-03178-f004:**
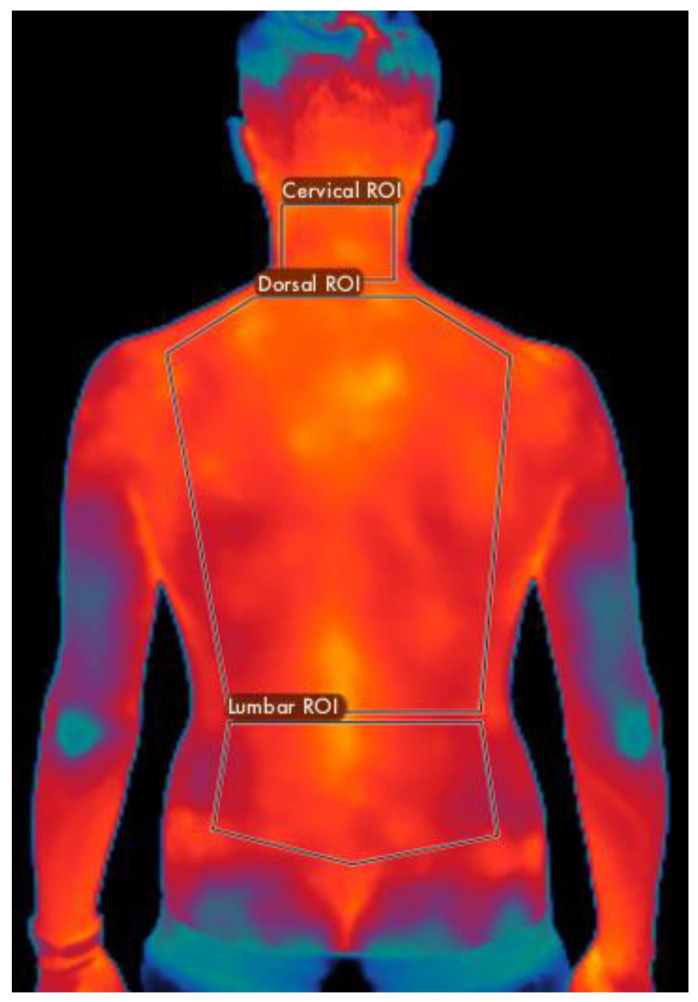
Identification of the cervical, dorsal, and lumbar ROI with FLIR Thermal Studio Pro.

**Figure 5 jcm-13-03178-f005:**
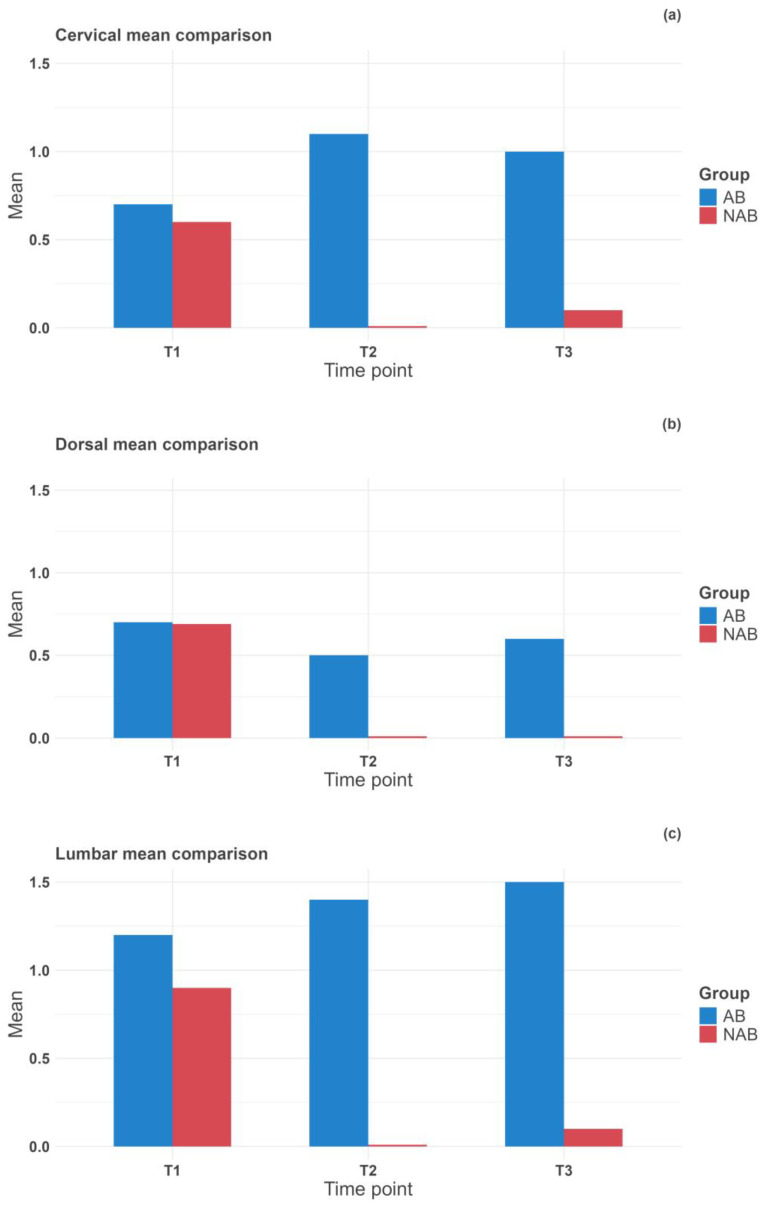
Boxplots representing the temperature differential differences of the cervical (**a**), dorsal (**b**), and lumbar (**c**) areas between the AB (blue) and the NAB (red) groups.

**Figure 6 jcm-13-03178-f006:**
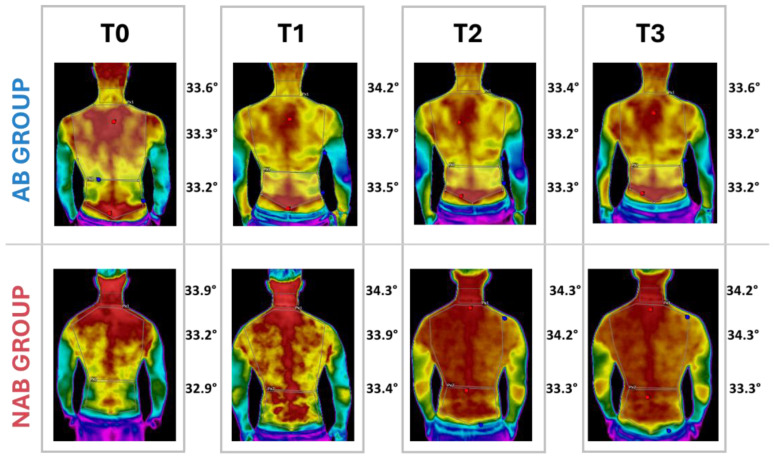
Temperature changes in the AB and NAB groups at different time points.

**Table 1 jcm-13-03178-t001:** Anthropometric measurements of the sample.

Measures	Mean ± SD	
	Men	Women
Age (years)	34.50 ± 3.40	35.13 ± 3.15
Height (cm)	176.34 ± 5.46	164.98 ± 7.25
Weight (kg)	83.29 ± 11.82	67.56 ± 9.03

**Table 2 jcm-13-03178-t002:** Mean temperature for each back area in the AB group and NAB group.

Measures	Mean ± SD		
	Cervical	Dorsal	Lumbar
T0 NAB	33.8 ± 0.6	33.2 ± 1.0	32.4 ± 1.3
T0 AB	33.8 ± 0.6	33.0 ± 0.8	32.5 ± 1.2
T1 NAB	34.3 ± 0.8	33.9 ± 0.8	33.3 ± 1.1
T1 AB	34.6 ± 0.6	33.8 ± 0.5	33.7 ± 0.9
T2 NAB	34.4 ± 0.9	33.9 ± 0.8	33.3 ± 1.0
T2 AB	33.4 ± 1.1	33.3 ± 0.7	32.3 ± 1.1
T3 NAB	34.2 ± 1.0	33.8 ± 0.8	33.3 ± 1.0
T3 AB	33.6 ± 0.9	33.2 ± 0.8	32.2 ± 1.1

AB = active break group; NAB = no active break group.

**Table 3 jcm-13-03178-t003:** Post-hoc analysis with Benjamini–Hochberg *p*-value adjustment to assess the significance of specific interactions in the AB group and NAB group.

Group Comparison	Cervical	Dorsal	Lumbar
1 T0/T1 NAB	0.0119 *	0.00335 **	0.00236 **
2 T1/T2 NAB	0.988	0.988	0.956
3 T1/T3 NAB	0.493	0.949	0.93
4 T2/T3 NAB	0.493	0.949	0.93
5 T0/T1 AB	0.00826 **	0.00672 **	<0.001 ***
6 T1/T2 AB	<0.001 ***	0.0528	<0.001 ***
7 T1/T3 AB	<0.001 ***	0.0274 *	<0.001 ***
8 T2/T3 AB	0.493	0.949	0.776

AB = active break group; NAB = no active break group; * = *p*-value < 0.05; ** = *p*-value < 0.01; *** = *p*-value < 0.001.

**Table 4 jcm-13-03178-t004:** Mean temperature differential values for each back area in the AB group and NAB group.

Measures	Mean ± SD		
	Cervical	Dorsal	Lumbar
Δ_t0-1_ NAB	0.6 ± 0.8	0.7 ± 0.7	0.9 ± 0.8
Δ_t0-1_ AB	0.7 ± 0.4	0.7 ± 0.6	1.2 ± 0.8
Δ_t1-2_ NAB	0.0 ± 0.4	0.0 ± 0.4	0.0 ± 0.9
Δ_t1-2_ AB	1.1 ± 0.9	0.5 ± 0.6	1.4 ± 0.7
Δ_t1-3_ NAB	0.1 ± 0.6	0.0 ± 0.5	0.1 ± 0.6
Δ_t1-3_ AB	1.0 ± 0.8	0.6 ± 0.7	1.5 ± 0.8

AB = active break group; NAB = no active break group; Δ_t0-1_ = differential of the T0 and T1 temperatures; Δ_t1-2_ = differential of the T1 and T2 temperatures; Δ_t1-3_ = differential of the T1 and T3 temperatures.

**Table 5 jcm-13-03178-t005:** Post-hoc analysis to assess the significance of temperature differentials in the AB group and NAB group.

Group Comparison	Cervical	Dorsal	Lumbar
Δ_t0-1_-NAB/AB	0.353	0.74	0.144
Δ_t1-2_-NAB/AB	<0.001 ***	0.00311 **	<0.001 ***
Δ_t1-3_-NAB/AB	<0.001 ***	0.00149 **	<0.001 ***

AB = active break group; NAB = no active break group; Δ_t0-1_ = differential of the T0 and T1 temperatures; Δ_t1-2_ = differential of the T1 and T2 temperatures; Δ_t1-3_ = differential of the T1 and T3 temperatures; ** = *p*-value < 0.01; *** = *p*-value < 0.001.

## Data Availability

The datasets used and/or analyzed during the current study are available from the corresponding author upon reasonable request.

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
