# Peer review of "Active Breaks Reduce Back Overload during Prolonged Sitting: Ergonomic Analysis with Infrared Thermography"

_jcm, 2024, doi:10.3390/jcm13113178_

Round 1
Reviewer 1 Report
Comments and Suggestions for Authors
Federico Roggio and his colleagues aim to evaluate the thermal fluctuations in the back resulting from prolonged periods of sitting and investigate how active breaks can alleviate the strain on the back in office workers. This will be done through the use of infrared thermography (IRT) to analyse the back during both prolonged sitting and active breaks. The findings indicate that the skin temperature of the back significantly increased in both groups after 30 minutes of sitting. However, in the following measurements, the AB group exhibited a drop-in temperature, but the NAB group maintained a high temperature. There were no statistically significant differences between the Δt0-1 of the NAB and AB. However, the Δt1-2 and Δt1-3 of the NAB and AB exhibited statistically significant differences for all back areas.
The paper is well written and the systematic analysis of 57 office workers using IRT in particular is interesting. Although the authors fail to introduce some basic concepts in the introduction section, the results presented is important and an addition to the existing literature. I recommend the paper for publication after addressing some of my major concerns.
Since the results presented based on the small sample size, the authors should avoid making strong statements on this aspect or otherwise it should be validated in a more robust way.
The authors should also ensure that the results presented is generalizable across different populations. Overfitting to specific populations can limit the utility of the results presented in real-world applications.
The limitation of the study should be mentioned in the abstract/conclusion of the manuscript.
The overall methodology adopted in the paper should be depicted as flow diagram to ease the readers.
The paper is lack of detailed summary of the advantages and disadvantages of the study presented.
The conclusion section must be rewritten to be more informative and unveil the most beneficial outcomes.
Author Response
Federico Roggio and his colleagues aim to evaluate the thermal fluctuations in the back resulting from prolonged periods of sitting and investigate how active breaks can alleviate the strain on the back in office workers. This will be done through the use of infrared thermography (IRT) to analyse the back during both prolonged sitting and active breaks. The findings indicate that the skin temperature of the back significantly increased in both groups after 30 minutes of sitting. However, in the following measurements, the AB group exhibited a drop-in temperature, but the NAB group maintained a high temperature. There were no statistically significant differences between the Δt0-1 of the NAB and AB. However, the Δt1-2 and Δt1-3 of the NAB and AB exhibited statistically significant differences for all back areas.
Reply: Thank you for your comments, we are glad to the reviewer opinions on our manuscript.
The paper is well written and the systematic analysis of 57 office workers using IRT is interesting. Although the authors fail to introduce some basic concepts in the introduction section, the results presented is important and an addition to the existing literature. I recommend the paper for publication after addressing some of my major concerns.
Reply: Thank you for the positive comment. We believe that after the comments of the reviewer 1 we improved the quality of our manuscript.
Since the results presented based on the small sample size, the authors should avoid making strong statements on this aspect or otherwise it should be validated in a more robust way.
Reply: We apologize if we speculated on the results. We revised the manuscript to avoid strong statements. We included in the limitations of the study the problem of the small sample size.
The authors should also ensure that the results presented is generalizable across different populations. Overfitting to specific populations can limit the utility of the results presented in real-world applications.
Reply: Thank you for this suggestion. We only discussed the office workers while other occupations, such as teachers or drivers, are exposed to experience musculoskeletal pain. Therefore, we included in the last part of the introduction more information on the presence of pain among workers with a sedentary behavior. We further mentioned different populations in the discussion and pointed out the validity of this approach for other sedentary workers in lines 87-93, 327-333.
The limitation of the study should be mentioned in the abstract/conclusion of the manuscript.
Reply: We apologize for this oversight. We clearly stated the limitations of the study in the discussion, abstract and conclusion.
The overall methodology adopted in the paper should be depicted as flow diagram to ease the readers.
Reply: Based on the reviewer suggestions, we have added a flow diagram to clarify the methodology, Figure 1. Thank you.
The paper is lack of detailed summary of the advantages and disadvantages of the study presented.
Reply: We would like to thank the reviewer for highlighting this issue. We have revised our manuscript to include the advantages and the limitations of this study in lines 306-344.
The conclusion section must be rewritten to be more informative and unveil the most beneficial outcomes.
Reply: Thank you for this comment. We took the opportunity to revise the conclusions and give a more straightforward information on the findings of the study.

Reviewer 2 Report
Comments and Suggestions for Authors
The study “Active breaks reduce back overload during prolonged sitting: ergonomic analysis with infrared thermography” investigated the effect of active breaks on back overload by measuring the skin temperature of the back via infrared thermography. The aim of the study is very clear, the study method is correct and the manuscript is well written and clarified. The cited references are sufficient.
There are some small problems about the manuscript.
1. Materials and methods. The clothes of the subjects worn should be clarified, since clothing made a great difference on skin temperature. Another, the skin temperature of the subjects were measured by the infrared thermography naked or with clothes on, should be given clearly.
2. Table 1. The numbers in the table should be revised. Decimal points should used not commas.
3. Figure 4. In the bar chart, the time points of the first bars should be T1 not T0.
4. Line 184. “Mean temperature differential values for each back area in the 183 AB and NAB are reported in Table 3, Figure 3”. Here should be Figure 4 not Figure 3.
Author Response
The study “Active breaks reduce back overload during prolonged sitting: ergonomic analysis with infrared thermography” investigated the effect of active breaks on back overload by measuring the skin temperature of the back via infrared thermography. The aim of the study is very clear, the study method is correct and the manuscript is well written and clarified. The cited references are sufficient.
There are some small problems about the manuscript.
- Materials and methods. The clothes of the subjects worn should be clarified, since clothing made a great difference on skin temperature. Another, the skin temperature of the subjects were measured by the infrared thermography naked or with clothes on, should be given clearly.
Reply: The study participants were wearing a loose-fitting T-shirt during the whole time of the study. When we were taking the measurement, we asked them to remove their upper body clothing to collect infrared thermography data. We added this part in the materials and methods in line 125-125. Thank you for highlighting this issue.
- Table 1. The numbers in the table should be revised. Decimal points should used not commas.
Reply: Thank you for the suggestion. We have replaced all commas with periods in table 2.
- Figure 4. In the bar chart, the time points of the first bars should be T1 not T0.
Reply: We have fixed the figure, thank you.
- Line 184. “Mean temperature differential values for each back area in the 183 AB and NAB are reported in Table 3, Figure 3”. Here should be Figure 4 not Figure 3.
Reply: We added tables and figures following the suggestions of all the reviewers. Consequently, we updated the numbering as well. Thanks for your support.

Reviewer 3 Report
Comments and Suggestions for Authors
Dear Authors,
Thank you for your submission.
I have enjoyed reviewing your paper. Overall, it is informative, well-written, and addresses an important topic of interest.
I have few suggestions for you to consider. Please do the following:
Abstract
Keywords: it is recommended to arrange keywords in an alphabetical order.
Materials and Methods / Physical activity protocol
lines 107-109: It would be really nice to add a figure with illustrative pictures of those really helpful and useful exercises for readers to better understand and hopefully apply at their workplace.
Results
I suggest putting participants demographics in a table instead of text (table will provide an easy-to-understand visual representation of demographics at baseline).
Discussion
line 275: can you elaborate more on physical activity and ergonomic interventions and support that with more references (i.e., provide examples of recommended workstations setup, workstation set-up and self-assessment checklist per recommended guidelines, if any, etc.)
I will be happy to review the revised manuscript.
Best wishes,
Author Response
Dear Authors, Thank you for your submission. I have enjoyed reviewing your paper. Overall, it is informative, well-written, and addresses an important topic of interest.
Reply: We would like to thank reviewer 2 for this positive comment.
I have few suggestions for you to consider. Please do the following:
Abstract Keywords: it is recommended to arrange keywords in an alphabetical order.
Reply: Thank you for this suggestion. We have rearranged the keywords in alphabetical order.
Materials and Methods / Physical activity protocol
lines 107-109: It would be really nice to add a figure with illustrative pictures of those really helpful and useful exercises for readers to better understand and hopefully apply at their workplace.
Reply: We agree with the reviewer on this issue. We added the Figure 2 that represent the exercises.
Results - I suggest putting participants demographics in a table instead of text (table will provide an easy-to-understand visual representation of demographics at baseline).
Reply: Sorry for this oversight, we included the table with the anthropometric measurements; thank you.
Discussion - line 275: can you elaborate more on physical activity and ergonomic interventions and support that with more references (i.e., provide examples of recommended workstations setup, workstation set-up and self-assessment checklist per recommended guidelines, if any, etc.)
Reply: Thank you for this suggestion. We included more information on the validity of active breaks in the work environment in the discussion section at lines 302-312, 324-330.
I will be happy to review the revised manuscript.
Reply: We hope the reviewer will like the changes of the manuscript after the revisions.

Reviewer 4 Report
Comments and Suggestions for Authors
Dear authors, your work is devoted to a very interesting topic, the problem of physical inactivity. The study used a comparative experiment. The comparison was between two groups, a group that took breaks between long periods of sitting and a group that did not. During physical inactivity, the temperature of the subjects' backs was assessed using an infrared thermal imager. The problem of physical inactivity is of interest, however, the shortcomings of the study should be noted:
1) The idea of the work is that body surface temperature is related to muscle tension. The article does not indicate how this relates to the health of the subjects? Why is high temperature bad and low temperature good?
2) The “Introduction” section should end with the research hypothesis. The hypothesis is not formulated, so the research problem is not clear.
3) It is clear from the study that physical exercise at 30-minute intervals reduces the surface temperature of the subjects’ backs, but it is not clear how this relates to the health of the subjects.
4) The graphs in Figure 3 are loosely related to the purpose of the study.
5) The authors should measure the tone of these muscles along with measuring body temperature.
6) The reason for the increase in back temperature may depend on various factors, not only physical inactivity. Therefore, the study should be supplemented with methods that prove what an increase in back temperature leads to.
7) It is clear that sitting in one place for a long time without moving is harmful; this does not require much research. This is not a scientific discovery. The conclusion of the work is self-evident.
Author Response
Dear authors, your work is devoted to a very interesting topic, the problem of physical inactivity. The study used a comparative experiment. The comparison was between two groups, a group that took breaks between long periods of sitting and a group that did not. During physical inactivity, the temperature of the subjects' backs was assessed using an infrared thermal imager. The problem of physical inactivity is of interest; however, the shortcomings of the study should be noted:
1) The idea of the work is that body surface temperature is related to muscle tension. The article does not indicate how this relates to the health of the subjects? Why is high temperature bad and low temperature good?
Reply: We agree with the reviewer that this topic needs to be clarified. We have added lines 56-61 in the introduction section that evidence research that clarify this issue. In our study, we found an increase in back temperature compared to baseline (T0) in subjects who performed only prolonged sitting. In contrast, subjects who practiced the active breaks protocol maintained a lower temperature similar to the first measurement (T0) after performing the exercises. Thus, there is no ideal temperature range for all people, but we do know that an increase in temperature from baseline may be given by functional overload.
2) The “Introduction” section should end with the research hypothesis. The hypothesis is not formulated, so the research problem is not clear.
Reply: Thank you for this suggestion. We realized that our research hypothesis was not very clear in the text. Consequently, we improved the introduction section (lines 98-101) by explaining the research hypothesis in more details as indicated by the reviewers.
- It is clear from the study that physical exercise at 30-minute intervals reduces the surface temperature of the subjects’ backs, but it is not clear how this relates to the health of the subjects.
Reply: Thank you for your comment. We want to highlight that the decrease in surface temperature, returning to near baseline levels after the exercise protocol, could be an important finding for health. The literature shows that an increase in temperature can lead to muscle fatigue (Machado ÁS, da Silva W, Priego-Quesada JI, Carpes FP. Can infrared thermography serve as an alternative to assess cumulative fatigue in women? J Therm Biol. 2023 Jul; 115:103612. doi: 10.1016/j.jtherbio.2023.103612. Epub 2023 Jun 22. PMID: 37379651). Therefore, a decrease in temperature, staying near to baseline, could indicate less muscle fatigue and thus less discomfort.
4) The graphs in Figure 3 are loosely related to the purpose of the study.
Reply: We agree with this comment. Following the reviewer's suggestion, we have removed Figure 3. Thank you.
5) The authors should measure the tone of these muscles along with measuring body temperature.
Reply: The suggestion raised by the reviewer is interesting and stimulating for new research. We only measured skin temperature by infrared thermography because we wanted to employ a method which is quick, easy, and feasible for workplace intervention. However, we believe that future studies should relate skin temperature to other measurements of muscle mechanical properties such as tone or stiffness. Hence, to follow all the reviewer’s suggestion we updated the limitation and future studies parts of our discussion. Thank you for your comment.
6) The reason for the increase in back temperature may depend on various factors, not only physical inactivity. Therefore, the study should be supplemented with methods that prove what an increase in back temperature leads to.
Reply: Thank you for this suggestion, we believe that there is still much to explore in the field of infrared thermography. For this reason, we decided to use only infrared thermography considering that it is an easy, quick, non-invasive, and reliable tool to use. These features make it suitable for frequent and detailed analyses. Also, in our study methods, we employed the TISEM checklist. Following these guidelines allows to consider and control many factors that might affect skin temperature, ensuring the accuracy and reliability of the measurements. The TISEM checklist was crucial to prevent any interference that could affect the results of the measurements taken with infrared thermography.
Ismael Fernández-Cuevas, Joao Carlos Bouzas Marins, Javier Arnáiz Lastras, Pedro María Gómez Carmona, Sergio Piñonosa Cano, Miguel Ángel García-Concepción, Manuel Sillero-Quintana, Classification of factors influencing the use of infrared thermography in humans: A review, Infrared Physics & Technology, Volume 71,2015, Pages 28-55, ISSN 1350-4495.
7) It is clear that sitting in one place for a long time without moving is harmful; this does not require much research. This is not a scientific discovery. The conclusion of the work is self-evident.
Reply: The scientific literature is indeed full of research showing the harmful effects of inactivity. However, our study implemented a specific protocol of active breaks aimed at reducing back overload that is feasible in a work setting. This type of intervention is particularly relevant because it can be applied directly in the workplace, improving employee wellness, and reducing the risks associated with sedentariness. We believe this topic can benefit from new data to further support corporate welfare initiatives and promote a healthier work environment.

Reviewer 5 Report
Comments and Suggestions for Authors
This study offers valuable insights into the detrimental effects of prolonged sitting on back health among office workers and the potential benefits of incorporating active breaks into their routines. The following questions need to be addressed prior to publication.
1. What is the research hypothesis addressed in this research work?
2. For ergonomic analysis, thermal analysis alone is not sufficient. The authors can additionally use a seat pressure sensor or sEMG to draw a better inference.
3. What method was used to measure muscle activity and strain?
4. What were the key findings regarding changes in skin temperature over time for both groups? Atmospheric temperature also has a significant effect on the objects.
5. Were there statistically significant differences between the groups in terms of temperature changes? If so, when did these differences occur? What are the research findings?
6. What are the implications of the study's findings for workplace interventions?
7. How does the study suggest that infrared thermography (IRT) can be utilized in occupational health settings? Exposure to IR radiation is not good for health.
8. How does the study recommend using IRT as a monitoring tool for interventions targeting musculoskeletal health in office workers?
9. Any suggested post-intervention proposed in this work.
10. Revise the conclusion with research findings.
Comments on the Quality of English LanguageMinor editing of the English language is required
Author Response
health among office workers and the potential benefits of incorporating active breaks into their routines. The following questions need to be addressed prior to publication.
- What is the research hypothesis addressed in this research work?
Reply: Thank you for the comment. We realized that the research hypothesis was not clear. Therefore, we better explained this in the last part of the introduction in lines 93-96.
- For ergonomic analysis, thermal analysis alone is not sufficient. The authors can additionally use a seat pressure sensor or sEMG to draw a better inference.
Reply: Thank you for this comment. In our study, we evaluated participants using only infrared thermography, but we would like to point out that the use of infrared thermography is already an accepted and sufficiently effective method for analyzing thermal changes in the human body as reported in the following studies:
- Soranso DR, Minette LJ, Marçal M, Marins JCB, Schettino S, Lima RCA, Oliveira M. Thermography in ergonomic assessment: a study of wood processing industry workers. PeerJ. 2022 Sep 20;10:e13973. doi: 10.7717/peerj.13973
- Luximon A, Chao H, Goonetilleke RS and Luximon Y (2022) Theory and applications of InfraRed and thermal image analysis in ergonomics research. Front. Comput. Sci. 4:990290. doi: 10.3389/fcomp.2022.990290)
Since the infrared thermography is a relatively new tool in the human body assessment, we included more information in the introduction on the validity of IRT as a screening tool in ergonomics in lines 63-69.
- What method was used to measure muscle activity and strain?
Reply: Thank you for the comment. As described in the methods, the only evaluative means is Infrared Thermography since previous studies confirmed its applicability in the ergonomics field. However, we believe that further studies may be conducted to compare the results of the infrared thermography with other methods; therefore, we included this as limitation and future studies suggestions.
- What were the key findings regarding changes in skin temperature over time for both groups? Atmospheric temperature also has a significant effect on the objects.
Reply: We apologize if we were not enough clear in the results. In lines 192-193 we stated that these results refer to the temperature changes over time. In lines 193-200 there are the results of both groups. Furthermore, lines 269-270 discusses this result again. Concerning the atmospheric temperature, we specified in the methods section that we followed the TISEM checklist. The TISEM checklist indicates the methodology to be adopted for thermographic studies to reduce measurement bias, such as room temperature. We also pointed out that the room temperature was set at 23° ± 2 °C and its humidity at 50%.
- Were there statistically significant differences between the groups in terms of temperature changes? If so, when did these differences occur? What are the research findings?
Reply: We analyzed the change of temperature by conducting the temperature differential analysis. The results shows that the temperature changed between the two groups in the T1-T2 and T1-T3 intervals. Indeed, the NAB group's temperature remained steady, showing no change between the T1, T2, and T3 measurements. In contrast, the temperature in the AB group decreased after the exercise protocol. These results are present in Table 4 and we further discussed it from lines 274-275 onwards.
- What are the implications of the study's findings for workplace interventions?
Reply: Thank you for this comment as we noticed that we were superficial about the clinical aspects of this study. Given this, we have included a part where we suggest the validity of such exercises for those who work in a sedentary environment in lines 305-315.
- How does the study suggest that infrared thermography (IRT) can be utilized in occupational health settings? Exposure to IR radiation is not good for health.
Reply: We would like to emphasize that the IRT is not an emitter of harmful radiation, it’s a detector. The sensor can detect the infrared radiation emitted by a body in a wavelength that is between 750 nm and 1 mm. Therefore, its main scope is to detect the infrared radiation relative to the temperature. Since it is widely being employed in aiding cancer diagnoses, we are confident in suggesting IRT as risk-free and non-invasive evaluation method.
- Faust O., Rajendra Acharya U., Ng E.Y.K., Hong T.J., Yu W. Application of infrared thermography in computer aided diagnosis. Infrared Phys. Technol. 2014;66:160–175. doi: 10.1016/j.infrared.2014.06.001
- Ng E.Y.K. A review of thermography as promising non-invasive detection modality for breast tumor. Int. J. Sci. 2009;48:849–859. doi: 10.1016/j.ijthermalsci.2008.06.015;),
- How does the study recommend using IRT as a monitoring tool for interventions targeting musculoskeletal health in office workers?
Reply: IRT does not require anything on the worker’s body, the thermogram can be captured in 30 seconds and easily analyzed, without harmful effects. Therefore, we believe that in the environment of sedentary workers, it could support the preventive medicine to reduce the risk of musculoskeletal disorders. Thanks to this question, we have explained this point better in the manuscript in the discussion section in lines 321-324.
- Any suggested post-intervention proposed in this work.
Reply: Although we struggled to understand the reviewer’s comment, we believe that this suggestion meant to include further information on the applicability of exercises in a work environment. Therefore, we better specified the validity of specific exercises for sedentary workers in lines 327-333.
- Revise the conclusion with research findings.
Reply: We revised the conclusions following the suggestions of all the reviewers, thanks for your valuable support.

Round 2
Reviewer 1 Report
Comments and Suggestions for Authors
No further comments
Reviewer 5 Report
Comments and Suggestions for Authors
Congrats to the authors.